# Free-living human cells reconfigure their chromosomes in the evolution back to uni-cellularity

Jin Xu[1†], Xinxin Peng[1†], Yuxin Chen[2†], Yuezheng Zhang[1], Qin Ma[1,3], Liang Liang[1,3], Ava C Carter[4], Xuemei Lu[1,3,5*], Chung-I Wu[1,2,6*]

[1]Key Laboratory of Genomic and Precision Medicine, Beijing Institute of Genomics, Chinese Academy of Sciences, Beijing, China; [2]State Key Laboratory of Biocontrol, School of Life Sciences, Sun Yat-Sen University, Guangzhou, China; [3]University of Chinese Academy of Sciences, Beijing, China; [4]Center for Personal Dynamic Regulomes, Stanford University School of Medicine, Stanford, United States; [5]CAS Center for Excellence in Animal Evolution and Genetics, Chinese Academy of Sciences, Kunming, China; [6]Department of Ecology and Evolution, University of Chicago, Chicago, United States

**Abstract** Cells of multi-cellular organisms evolve toward uni-cellularity in the form of cancer and, if humans intervene, continue to evolve in cell culture. During this process, gene dosage relationships may evolve in novel ways to cope with the new environment and may regress back to the ancestral uni-cellular state. In this context, the evolution of sex chromosomes vis-a-vis autosomes is of particular interest. Here, we report the chromosomal evolution in ~ 600 cancer cell lines. Many of them jettisoned either Y or the inactive X; thus, free-living male and female cells converge by becoming 'de-sexualized'. Surprisingly, the active X often doubled, accompanied by the addition of one haploid complement of autosomes, leading to an X:A ratio of 2:3 from the extant ratio of 1:2. Theoretical modeling of the frequency distribution of X:A karyotypes suggests that the 2:3 ratio confers a higher fitness and may reflect aspects of sex chromosome evolution.
DOI: https://doi.org/10.7554/eLife.28070.001

*For correspondence:
luxm@big.ac.cn (XL);
ciwu@uchicago.edu (C-IW)

†These authors contributed equally to this work

Competing interests: The authors declare that no competing interests exist.

## Introduction

Genomes of multi-cellular organisms evolve to ensure the survival and reproduction of the whole organisms. With human interventions akin to domestication, hundreds of cell lines survive as free-living cells that are not organized into tissues, organs or individuals (*Alberts et al., 2002*). Evolution in such a quasi-unicellular state may be very different from the evolution as multi-cellular entities. Most cell lines are cancerous in origin but a few are derived from normal tissues (*Hayflick, 1998*). Regardless of their origin, they have all evolved characteristics for survival in the unicellular state that is distinct from their natural environments. Cell lines derived from cancer tissues are usually karyotypically less stable than normal cell lines (*Lengauer et al., 1997*). While this instability may impose a cost, it also permits cancer cell lines to evolve new karyotypes, including polyploidy, more readily than normal cell lines could.

Tumorigenesis has been increasingly viewed as a process of evolution, rather than merely pathological conditions (*Nowell, 1976*; *Merlo et al., 2006*). This 'ultra-microevolutionary process' is subjected to similar rules including mutation, genetic drift, migration and selection that govern organismal evolution (*Wu et al., 2016*). While this process usually ends when the organism dies, cell lines in the cultured state will continue to evolve. Much like the diversity unleashed by

**eLife digest** Multicellular life relies on a group of cells working together for a common interest. To study these cells, researchers take them out of the organism and grow them in the laboratory. Instead of growing as part of organs and tissues, the cells normally have a free-living lifestyle. Because multicellular life evolved from single-celled organisms, laboratory-grown cells can be considered as life forms that are evolving backward from a multicellular to a single-celled existence.

Normally, the cells that make up most of the tissues in the human body have 22 pairs of chromosomes known as autosomes and a pair of sex chromosomes. The cells of women have two X sex chromosomes, one of which is inactive, while those of men have one X and one Y chromosome. However, free-living single cells do not need to distinguish between male and female cells.

Xu, Peng, Chen et al. have now studied the chromosomes of cancer cells taken from over 600 people and grown in the laboratory. As the cells evolved in response to their free-living lifestyle, they became 'de-sexualized'; male cells lost their Y chromosome, while female cells abandoned their inactive X chromosome. The cells then evolved toward a new state in which they possessed two active X chromosomes and three sets of autosomes. This new configuration suggests that the current X chromosome to autosome ratio may not be optimal for fitness and hence sheds some light on how mammalian sex chromosomes evolved.

It is currently thought that as cancerous tumors grow, their cells evolve to favor their own interests over the common interests of the rest of the organism. In this way, they develop characteristics more like those of single cells. Further research is therefore needed to investigate whether changes occur to the chromosomes of cancer cells growing within the body, and whether this gives them an advantage over normal cells.
DOI: https://doi.org/10.7554/eLife.28070.002

domestication, cultured cell lines, which can be considered 'domesticated', may be informative about the evolutionary potentials at the cellular level.

In this quasi-unicellular state, gene dosage has been observed to change extensively as polyploidy, aneuploidy (full or partial) and various copy number variations (CNVs) are common in cancer cell lines (*Roschke et al., 2003*). Since these cell lines are derived from somatic tissues of men or women (referred to as male and female cells, for simplicity), they should be different in their sex chromosomes in relation to the autosomes (A's). Nevertheless, the possibility of separate evolutionary paths has not been raised before. Somatic cells have an inactive X chromosome in females and a Y chromosome in males (*Charlesworth, 1991*). Since cell lines presumably do not need sexual characters, we ask how the X:A relationship might have evolved in both male and female cells. More generally, we ask whether the evolution in this relationship may shed light on the emergence of mammalian sex chromosomes and their subsequent evolution.

In this study, we analyze 620 cancer cell lines that have been genotyped using SNP arrays (*Greenman et al., 2010*). Among them, 279 are derived from female tissues and 341 from male tissues. We observed the elimination of the Y and the inactive X chromosome, followed by the evolution toward a new equilibrium with two active X chromosomes and 3 sets of autosomes (2X:3A). We discuss the implication of these findings for the evolution of sex chromosome, the transition between uni- and multi-cellularity and cancers biology.

## Results

### Convergent sex chromosome evolution between sexes

The most common form of genomic changes in cell lines is the loss of heterozygosity (LOH) when one of the two homologous chromosomes is eliminated (*Roschke et al., 2003*). We therefore examine single nucleotide polymorphisms (SNPs) across the 620 cell lines for occurrences of LOH on each autosome and the X chromosome. Male and female cell lines are separately analyzed.

*Figure 1A* shows the LOH frequency for each autosome (black dots) and the red dot represents the sex chromosomes (X in female and Y in male). For autosomes, the percentages of LOH are remarkably similar between sexes, with a correlation coefficient of 0.94 among 620 cell lines. There

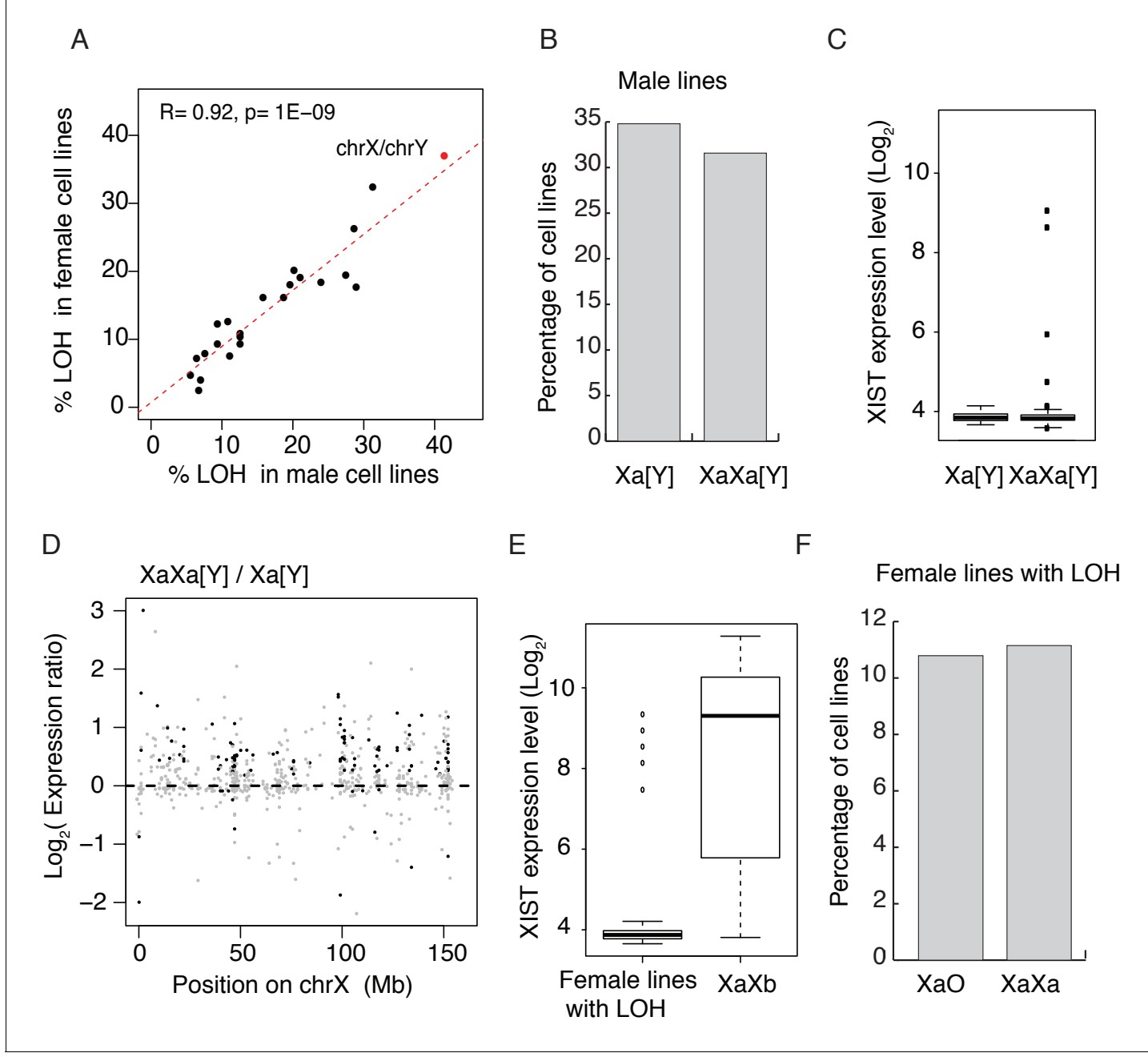

**Figure 1.** Convergence in sex chromosomes in culture human cells. (**A**) Percentage of lines with LOH (loss of heterozygosity). Each black dot represents an autosome and the red dot represents X and Y. LOH in male and female lines are separately displayed on the X and Y-axes. (**B**) Percentage of cell lines with either one (n = 119, 34.80% of male cell lines) or two Xa's (n = 108, 31.58% of male cell lines), cell lines with partial X's are not included. (**C**) Expression level of XIST in male cell lines, [Y] means with or without Y chromosome. (**D**) Expression ratios of X-linked genes between Xa[Y] and XaXa[Y] cell lines. Each grey dot represents a gene, and significant differences are indicated by black dots (t-test, p<0.05). (**E**) Expression level of XIST in female cell lines with or without LOH. Female lines with LOH have very low levels of XIST, suggesting all X's being active. In non-LOH (XaXb) lines, the expression of XIST indicates the presence of inactive X's. (**F**) Percentage of cell lines with either one (n = 30,10.79% of female cell lines) or two Xa's (n = 31, 11% of female lines) in female lines with LOH of whole X chromosome. Non-LOH lines are not used because of the uncertainty in the number of Xa's.

DOI: https://doi.org/10.7554/eLife.28070.003

The following figure supplement is available for figure 1:

**Figure supplement 1.** The frequency of chromosomes loss show negative correlation to their length.

DOI: https://doi.org/10.7554/eLife.28070.004

is a slight tendency for the smaller autosomes to have higher LOH rate (R =~−0.4, p=~0.046, *Figure 1—figure supplement 1*). The median percentage of LOH is about 13% for autosomes. However, the losses of X (37% in females) and Y (40% in males) stand out. Given its rank as the 7[th] largest chromosome, the X is not expected to be lost in more than 15% of cell lines, based on the regression analysis of *Figure 1A*. Since the expression from the X is not lost, we infer that it's the inactive X(or Xi) that is eliminated.

Female lines lose the inactive X (Xi) and male lines lose the Y chromosome at a higher rate than other chromosomes. The two sexes may thus be expected to converge toward having a single sex chromosome. Furthermore, given that spontaneous LOH is not infrequent and the loss cannot be regained, long-term cultures might evolve to complete LOH for sex chromosomes as well as autosomes. The genome-wide low rate of LOH suggests selection holding back such changes. The strong correlation between sexes further reflects a balance between the production and elimination of LOH's, likely involved natural selection.

A most unexpected finding is that, accompanying the loss of the Y or Xi, an extra X chromosome is often gained. *Figure 1B* shows approximately equal numbers of male cell lines with one or two X chromosomes (partial X aneuploidy not counted). This extra X is active because the inactivating XIST lncRNA is silenced in male cell lines (*Figure 1C*), consistent with previous findings (*Guttenbach et al., 1995*). XIST does not become activated in free-living cells that do not already express this. The expression of X-linked genes is higher in those male lines with two X's than in those with one X and the up-regulation occurs along the length of the X chromosome (*Figure 1D*).

The pattern is more complex in female lines which, in their original state, contain an Xa and an Xi, the latter expressing XIST (*Chow et al., 2005*; *Plath et al., 2002*; *Ng et al., 2007*). We use only female lines that show LOH of the whole X chromosome (~37% of female lines) in counting Xa's for the following reason. In order to count active Xa's, we require the absence of XIST expression in the line such that all X's can be assumed active. *Figure 1E* shows that female lines with LOH indeed rarely express XIST, presumably because LOH lines that survive lost the inactive X and kept the active Xa. In contrast, non-LOH lines tend to express XIST, thus obscuring the counting of Xa's. Of the 103 LOH female lines, 30 lines have single Xa and 31 lines have whole extra X's as shown in *Figure 1F*. Much like male lines of *Figure 1B*, *Figure 1F* also shows roughly half of female lines to have gained an extra Xa.

Cancer cell lines usually have high rates of aneuploidy and could be heterogeneous within a given line, thus making its status difficult to assess. To assess the level of within-line heterogeneity, we chose two representative cell lines to count the X chromosomes in individual cells using fluorescent *in situ* hybridization (FISH). The two lines are A549 (a male cell line from adenocarcinomic alveolar basal epithelium) and HeLa (a female cervical cancer cell line). Neither line expresses XIST (*Supplementary file 1*), suggesting that all X chromosomes are active. *Figure 2A–B* shows results from individual A549 and HeLa cells with two and three X's. *Figure 2C–D* shows the X karyotype distributions. While there is a modest degree of heterogeneity within each line, almost all cells have two or more active X chromosomes. While labor intensity of assays and cell availability limited our sample size, we nevertheless can conclude that within-cell line heterogeneity does not seem to undermine our conclusions.

## Evolution toward a new X:A expression ratio ($E_{X/A}$)

With an extra copy of the active X, the 'expression phenotype' is expected to change. The ratio of the median gene expression on the X to that on the autosomes($E_{X/A}$) is of particular interest. $E_{X/A}$ has been reported to be around 0.5 ~ 0.8 for normal mammalian tissues (*Xiong et al., 2010*; *Deng et al., 2011*; *Kharchenko et al., 2011*). We assayed $E_{X/A}$ by separating lines derived from cancerous and normal tissues. *Figure 3A* shows that $E_{X/A}$ distributions center on ~ 0.84 in normal cell lines and on one in cancerous cell lines. Given the controversy in the assay of $E_{X/A}$, we also varied the threshold for counting expressed transcripts (see Materials and methods). By varying the threshold (*Figure 3B*), $E_{X/A}$ ranges from 0.78 to 1.05 in normal cell lines but is consistently higher by approximately 15% in cancer cell lines. The same pattern is seen in the RNA-seq data (*Figure 3—figure supplement 1*).

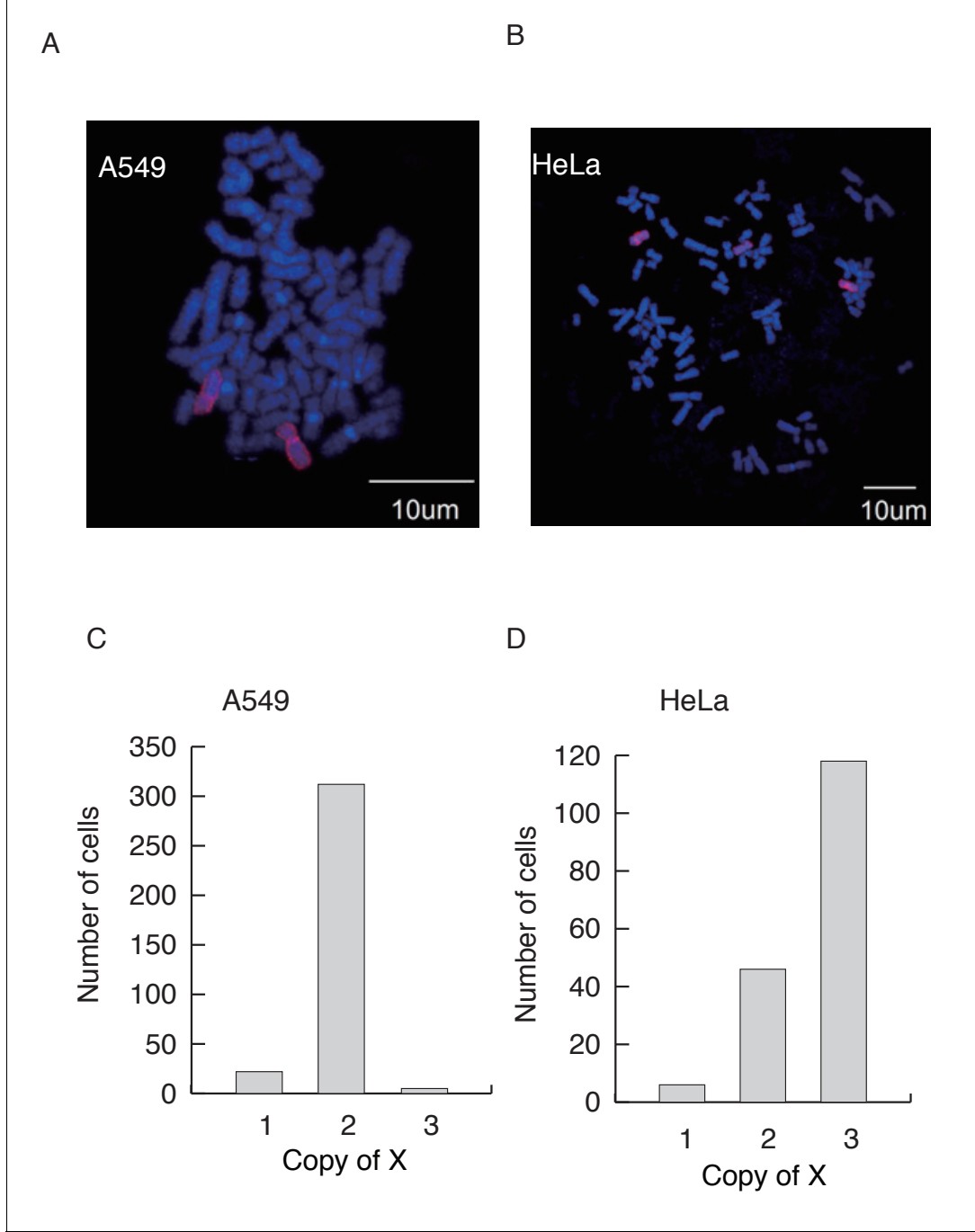

**Figure 2.** X chromosomes in indivudial cells. (A–B) Representative images of X chromosome FISH in the A549 cell line (A) with two Xs and HeLa (B) with three Xs. DNA is stained with DAPI (blue), and the X chromosome is labeled with Cy3 (red). (C–D) The distribution of the copy number of X's among cells from A549 (n = 343) and HeLa (n = 170).

DOI: https://doi.org/10.7554/eLife.28070.005

## The concerted evolution of autosomes as a set

While sex chromosomes evolve, autosomes should also evolve. Since the generation of aneuploidy may happen independently for each autosome, a key question is whether selection operates on the autosomes as a set. Does natural selection favor cells that have full sets of autosomes?

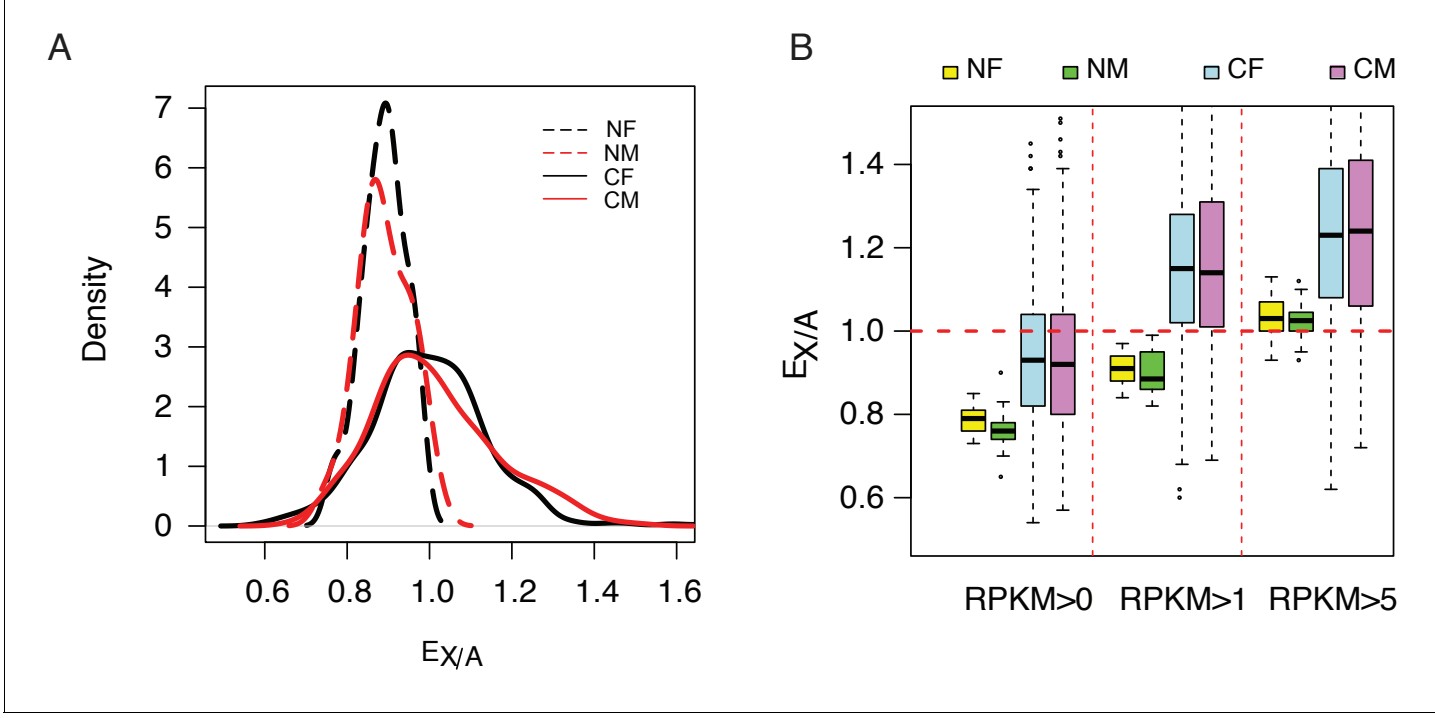

**Figure 3.** Increasing of expression ratio of X versus autosome ($E_{X/A}$). (**A**) $E_{X/A}$ distributions among normal (N) and cancer (C) cell lines. NF and NM (or CF and CM) designate normal (or cancer) female and male lines. $E_{X/A}$ in cancer cell lines become larger than those of the normal cell lines. Note that the expression in normal cell lines is narrowly distributed and is close to that of the normal tissue when compared. Although the numbers of NF and NM lines are much smaller than CF and CM lines (17 and 24 vs. 279 and 341), their $E_{X/A}$ distributions are much tighter than in cancer cell lines. The actual counts correspond to kernel density are given in *Figure 3—figure supplement 2*. (**B**) $E_{X/A}$ ratio in CF, CM, NF and NM lines with filtering with three different cutoffs (see Materials and methods). $E_{X/A}$ ratios are consistently higher in CF and CM lines than in NF and NM lines.

DOI: https://doi.org/10.7554/eLife.28070.006

The following figure supplements are available for figure 3:

**Figure supplement 1.** $E_{X/A}$ ratio in cancerous and normal cell lines by RNA-seq.

DOI: https://doi.org/10.7554/eLife.28070.007

**Figure supplement 2.** The frequency spectrum of $E_{X/A}$ in male and female cancerous cell lines compared to normal male and female cell lines.

DOI: https://doi.org/10.7554/eLife.28070.008

*Figure 4A* shows the distribution of chromosome number across the 620 cell lines we studied. Apparently, cancerous cell lines acquire autosomes during evolution. The distribution of ploidy (n = 22) number shows peaks at 2 and 3, indicates that many cell lines appear to be in transition between full diploidy and triploidy of 44 and 66 autosomes. Similarly, the majority of sublines of HeLa cells we examined have 55–75 chromosomes centering about the triploid count of 69 (*Figure 4—figure supplement 1*). Indeed, autosomes appear to exist as a full complement with n = 22. Although autosomes may evolve as a set, cells most likely add one autosome at a time. It is hence desirable to track each chromosome individually. Single cells were individually isolated from a HeLa cell line and subsequently grown to a sub-line of $10^6$ cells. We subjected six such sub-lines to whole genome sequencing such that each chromosome could be tracked individually. Smaller chromosomes are indeed more erratic in their numbers in cell lines. Only the largest 14 chromosomes (13 autosomes and X), which together account for ~75% of the genome, are used to test the convergence of autosomes. The cutoff is based on the observation that chromosome 13 is the largest autosome yielding viable trisomic new-borns (*Taylor, 1968*; *Patterson, 2009*; *Kleijer et al., 2006*). We reason that, if whole organisms can survive trisomy, the fitness consequence of the particular aneuploidy would probably be very small at the cellular level.

In all 6 lines, each of the 13 autosomes has 2–4 copies, ranging from an average of 2.62 to 3.23 (*Supplementary file 2*). If each autosome behaves independently, the number of autosomes that increase by x copies (x = 0, 1, 2 etc.) should follow a Poisson distribution with a mean of λ. Two

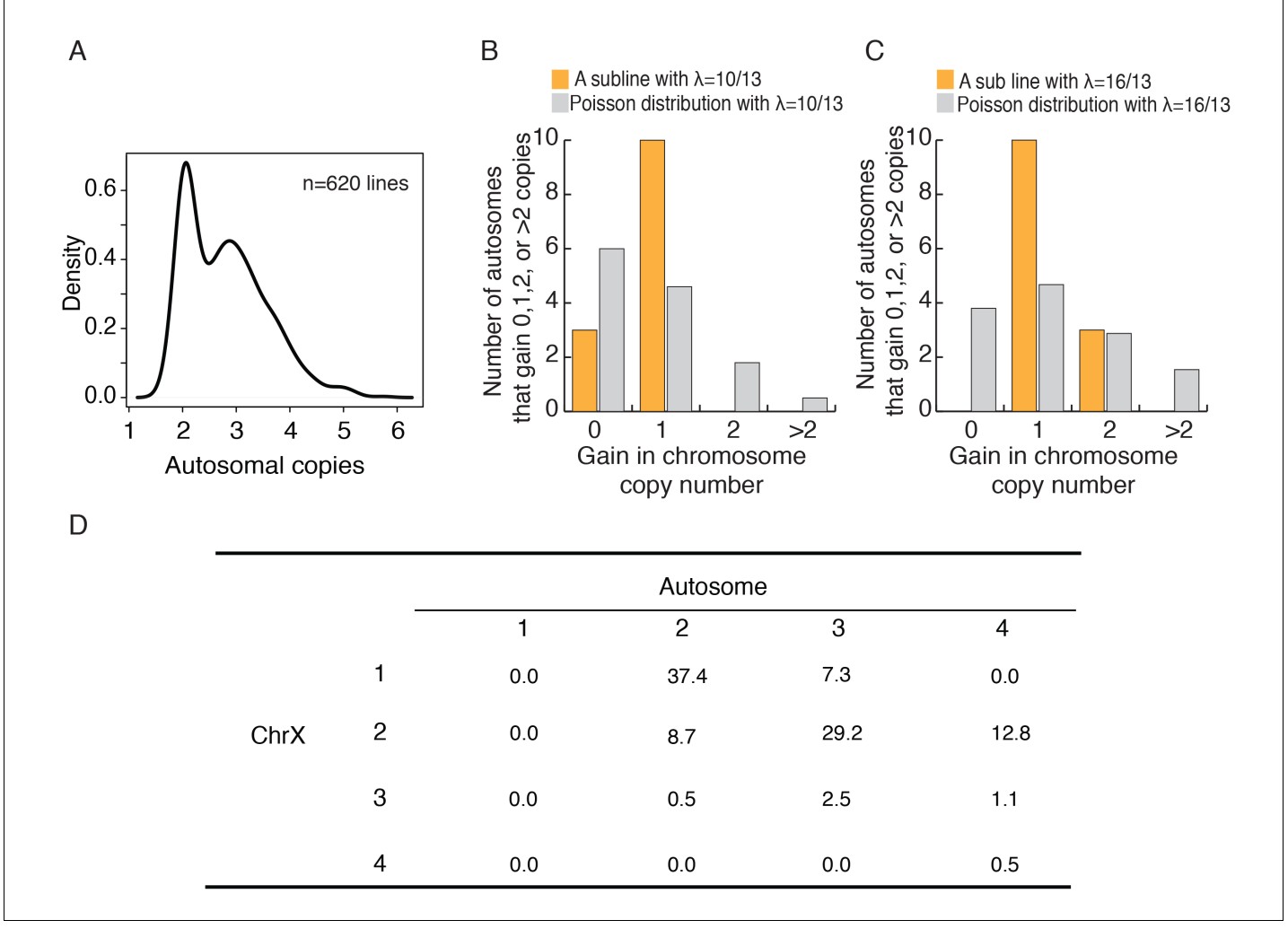

**Figure 4.** Autosomes change in a cohesive manner and coevolution of X and A. (**A**) The density plot of autosome copy number among 620 cell lines shows peaks at 2 and 3 per autosome. (**B-C**) The observed distributions of gain in copy number among autosomes in two HeLa sublines. The expected Poisson distributions are also given for sublines with different means ($\lambda$ = 10/13, 16/13; see text). (**D**) The percentages of C(Xa:A) types among the 620 cell lines.

DOI: https://doi.org/10.7554/eLife.28070.009

The following figure supplements are available for figure 4:

**Figure supplement 1.** Both the ancestral and sub-clonal HeLa population have 55–75 chromosomes centering around the triploid count of 69.
DOI: https://doi.org/10.7554/eLife.28070.010

**Figure supplement 2.** The frequency spectrum of C(Xa:A) across all male and female cancerous cell lines.
DOI: https://doi.org/10.7554/eLife.28070.011

different lines, with $\lambda$ = 10/13 and $\lambda$ = 16/13, are shown in *Figure 4B and C*. In the former, all cells have x = 0 or x = 1 and, in the latter, all cells have x = 1 or x = 2 (*Supplementary file 2*). The data suggest that each autosome increases by one copy and only after all of the 13 autosomes have gained an extra copy do further increases continue. *Figure 4—figure supplement 1* shows the composite distribution of the five lines with $\lambda$ < 1. The pattern, like that of *Figure 4B*, is statistically significant (p=0.0021 by the $\chi^2$ test) with an excess at x = 1. These results suggest that the larger autosomes evolve cohesively as a set. With autosomes evolving as a cohesive unit, X:A can be represented by whole numbers of 1:2, 2:3 etc.

## Evolution of the C(Xa:A) ratio underlying $E_{X/A}$

We now summarize the evolution of cell lines by their C(Xa:A) genotypes. C(Xa:A) is the number of active X chromosomes and the ploidy number of autosomes (in multiples of 22) and is equal to C(1:2) in normal cells. For the purpose of counting active Xa's, data from most male lines are usable. For female lines, only data from the LOH lines of the X can be used. Between the two sexes, C(Xa:A) distributions are very similar and the combined distribution is used in the analysis (*Figure 4—figure supplement 2*).

Shown in *Figure 4D*, most lines have the C(1:2) or C(2:3) genotype which together account for 2/3 of the lines. Given that C(1:2) is the starting genotype, its common occurrence at 37.4% is not surprising. The high frequency of C(2:3), however, is unexpected. To reach C(2:3) from the starting point of C(1:2), cells should evolve to either C(2:2) or C(1:3) first, but neither genotype is commonly seen in these cells lines. In contrast, C(2:3) at 29.2% is the second most common genotype. If we include the two genotypes, C(2:4) and C(3:3), that are derivatives of C(2:3), this inclusive C(2:3) cluster is the most common genotype. The model p the next section helps to interpret the observation.

## A model for the evolution of free-living cells

The pathways of chromosomal evolution can be diagrammed as a series steps in *Figure 5A*. Each node represents a C(Xa:A) genotype, the abundance of which is reflected in the size of the node. Thicker arrows indicate faster transitions which add/delete one X while the thinner arrow denotes the slower transition of adding/deleting the whole set of autosomes. The fitness of each genotype, W, is assumed to be determined by the Xa/A ratio. In general, one would expect the wild type ($W_1$) to be the fittest genotype and we particularly wish to know whether that is indeed the case here.

We first model the evolution under strict neutrality where all nodes have the same fitness. For simplicity, genotypes are grouped into 3 clusters centering around the 3 dominant genotypes, C(1:2), C(2:2) and C(2:3), the frequencies of which are $x_1$, $x_2$ and $x_3$, respectively. Each cluster consists of the dominant genotype as well as the less common ones adjacent to it (see *Figure 5A*). For instance, $x_2$ is the sum of the frequencies of C(2:2) and C(3:2) and $x_1$ is those of C(1:2), C(1:1) and half of C(1:3). The frequency of the last one, being adjacent to both C(1:2) and C(2:3), is split between the two clusters. Tallying up the numbers in *Figure 4D*, we obtain $x_1 = 0.41$, $x_2 = 0.092$ and $x_3 = 0.482$ with a total of 0.984, excluding the marginal genotypes. The analysis below can be expanded to account for each genotype separately. The transitions between clusters are defined as follows:

$$x_1(T) \underset{au}{\overset{u}{\rightleftharpoons}} x_2(T) \underset{bv}{\overset{v}{\rightleftharpoons}} x_3(T)$$

where u and v are the transition rates and $x_i(T)$ is the frequency of cluster i at time T. Let $X(T)$ be the vector of $[x_1(T), x_2(T), x_3(T)]$, expressed as

$$X(T) = X(0) \begin{bmatrix} 1-u & u & 0 \\ au & 1-au-v & v \\ 0 & bv & 1-bv \end{bmatrix}^T \tag{1}$$

When T >> 0,

$$[x_1(T), x_2(T), x_3(T)] \sim [ab, b, 1]/z \tag{2}$$

where z = ab + b+1. The genotype frequencies evolve toward the equilibrium, [ab, b, 1]/z, which depends on a and b, but not u and v. We posit that a > 1 and b > 1 because, as the chromosome number increases, the probability of chromosome gain/loss increases as well. By *Equation 2*, $x_1(T) > x_2(T) > x_3(T)$ when T >> 0. In short, the relative frequency should be in the descending order of C(1:2), C(2:2) and C(2:3) if there is no fitness difference among genotypes. This predicted inequality at T >> 0 is very different from the observed trend.

*Equation 2* assumes that cell lines have been evolving long enough to approach this equilibrium. A more appropriate representation should be X(T) where T reflects the time a cell line has been in culture. It is algebraically simpler if T is measured by the rate of chromosomal changes, u or v, rather

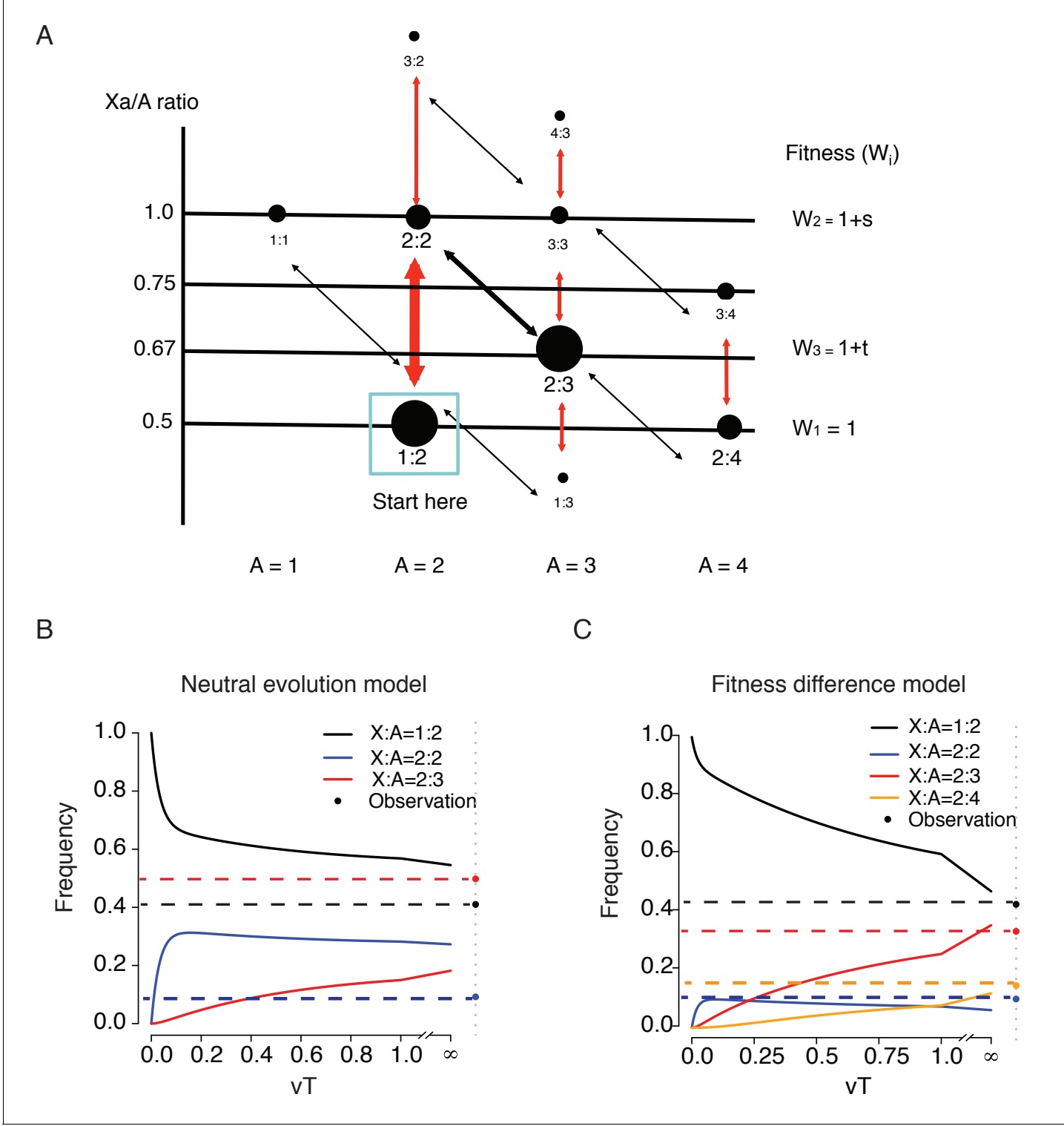

**Figure 5.** A model of karyotypic evolution driven by fitness differences. (**A**) Evolutionary pathways of chromosomal changes. Each node represents a karyotype C(Xa:A) and the size roughly corresponds to its frequency. Cell fitness is assumed to be a function of the Xa/A ratio, which is represented by the Y-axis. The four abundant karyotypes are shown by solid black circles. Red arrows indicate faster changes in X and black arrows indicate slower changes in autosome. Main transitions between the common karyotypes are indicated by thicker arrows. (**B**) Changes in the frequencies of the three key genotypes as a function of time (T, expressed in units of 1/v under fitness neutrality with all $W_i$'s = 1. The parameters for *Equations 1 and 2* are u = 10 v, a = 2 and b = 1.5. Both the theoretical trajectories and the observed values are given. The C(2:3) cluster ($x_3$) is far more common in the observation

*Figure 5 continued*

than in the neutral model. (**C**) Changes in 4 karyotypic frequencies under selection according to Equation S3 with s = −0,5 and t = 0.5. All other conditions are the same as above. Under selection, a reasonable agreement between the model and the observation can be obtained.

DOI: https://doi.org/10.7554/eLife.28070.012

than by the actual cell generation (*Equation 1*, *Figure 5B* and legends). We also assume u > v as u involves only the X but v involves the whole set of autosomes. With the initial condition of X(0) = [1,0,0], *Figure 5B* shows that the C(2:3) cluster approaches the equilibrium more slowly than the other two clusters. Therefore, the observed high frequency of the C(2:3) cluster ($x_3$ = 0.482 vs. $x_1$ = 0.41 and $x_2$ = 0.092) is incompatible with a neutrally evolving model of chromosome numbers. The discrepancy is true at all time points and is more pronounced at smaller T's.

Rejecting the neutral evolution model, we now incorporate fitness differences into *Figure 5A* with $W_1$ = 1 [for C(1:2) and C(2:4)], $W_2$ = 1 + s [for C(2:2)] and $W_3$ = 1 + t [for C(2:3)] where s and t can either be positive or negative. Here, we add a fourth genotype, C(2:4). In the supplement, we model 4 genotypes with $x_1 - x_4$ for the frequencies of C(1:2), C(2:2), C(2:3) and C(2:4) respectively. An expanded transition matrix is used to model selection, followed by a normalization step (*Supplementary file 3*, Equation S1). The solution in the form of X(T)=X(0) $M^T$ is given in *Supplementary file 3* (Equation S2) and the equilibrium X(T) is given in *Supplementary file 3* (Equation S3).

We are particularly interested in whether t > 0 in the 4-genotype model, that is, whether C(2:3) has a higher fitness than the wild type, C(1:2). We observe that [$x_1$, $x_2$, $x_3$, $x_4$] = [0.374, 0.087, 0,292, 0.128] where x3 = 0.292 is more than 3 times higher than $x_2$ = 0.087 and is close to $x_1$ = 0.374. Equation S3 shows that s < 0 is necessary for $x_2$ to be smaller than $x_3$, and t > 0 is necessary for $x_3$ to be close to $x_1$ (see Supplement). *Figure 5C* is an example in which s = −0.5 and t = 0.5. The equilibrium at T >> 0 is indeed close to the observed values.

In conclusion, it appears that the extant state in multicellular organisms of C(1:2) is not the fittest genotype for free-living mammalian cells. The observed genotypic distributions suggest that C(2:3) may have a higher fitness than the wild type, C(1:2).

## Discussion

Free-living mammalian cells like all living things speed up the evolution when the environment changes. The practice of cell culturing, however, is to slow down the evolution to preserve cell lines' usefulness as proxies for the source tissues. Nevertheless, changes are inevitable and the evolution of sex chromosomes is but one example. It should be noted that cell lines derived from cancerous tissues and normal tissues are different in one important aspect. Cell lines derived from normal tissues generally do not undergo karyotypic changes at an appreciable rate (*Shirley et al., 2012*; *Frazer et al., 2007*; *Pickrell et al., 2010*). They are therefore much less responsive to selection in cultured conditions that favor new karyotypes. Cancer cell lines, having been through more rounds of passages, have generally experienced stronger selection more frequently than normal cell lines.

Our observations suggest that the extant X:A relationship (C(1:2)) may not be optimal for free-living mammalian cells. The highest fitness peak, instead, appears to be closer to the karyotype of C(2:3) as free-living cells reproducibly evolve toward this new karyotype. The fitness peaks in free-living cells being different from that of the multi-cellular organisms is not unexpected. With many possible conflicts between individual cells and the community of cells (i.e., the organism), the interest of the community may lie in its ability to regulate the growth potential of its constituents. Free-living cells, on the other hand, are driven by selection to realize their individual proliferative capacity relative to other cells.

The convergence among these many cell lines to C(2:3) is unexpected in the context of cancer evolution. The TCGA project (reference) has shown that cancer evolution is a process of divergence, not convergence. Indeed, only two genes have been mutated in more than 10% of all cancer cases and tumors of the same tissue origin from two different patients may often share no mutated genes at all (*Wu et al., 2016*; *Kandoth et al., 2013*). Therefore, the karyotypic convergence reported here is rather unusual.

We note that C(2:3) toward which cultured cells evolved happens to be the smallest possible increase in the X/A ratio from C(1:2). The higher fitness of C(2:3) than C(1:2) in free-living cells may lend new clues to the debate about the evolution of mammalian sex chromosomes (*Kharchenko et al., 2011*; *Lin et al., 2012*). With X-inactivation, it has been suggested that $E_{X/A}$ could have been reduced, or even halved (*Xiong et al., 2010*; *Lin et al., 2012*). The debate is about whether, and by how much, $E_{X/A}$ might have increased in evolution. Our observation that free-living cells continue to evolve toward C(2:3) raised the possibility that the evolutionary increase in $E_{X/A}$ has not been complete, in comparison with the ancestral $E_{X/A}$.

Finally, this study of cancerous cell lines may also have medical implications. The common view that tumorigenesis is an evolutionary phenomenon posits that individual cells in tumors evolve to enhance self-interest (*Nowell, 1976*; *Merlo et al., 2006*; *Chen et al., 2015*; *Chen and He, 2016*). A corollary would be that tumorigenesis may have taken the first few steps toward uni-cellularity. This extended view is supported by many expression studies as well as the higher likelihood of obtaining cell lines from tumors than from normal tissues (*Hayflick, 1998*). An alternative view, posits that tumors remain multi-cellular in organization (*Almendro et al., 2013*). These different views have been critically examined recently (*Wu et al., 2016*). It is possible that cancer cells in vivo may have been gradually evolving toward a new optimum. In that case, cancer cells in men and women are converging in their sex chromosome evolution and become more efficient in proliferation in this new de-sexualized state.

## Materials and methods

### Chromosome number estimation of HeLa sub-lines

The processing of clonal expansion and whole genome sequencing of HeLa lines are described at Zhang et. al. (https://www.biorxiv.org/content/early/2017/10/05/193482). For each line, the copies of each chromosome are estimated according to the average sequencing depth by Control-FREEC, a tool for assessing copy number using next generation sequencing data (*Boeva et al., 2012*).

### Data collection

Three large-scale datasets were used in this study (*Greenman et al., 2010*; *Barretina et al., 2012*; *Cheung et al., 2010*).

Genome-wide SNP array data on cancer cell lines and a normal training set were downloaded from The Wellcome Trust Sanger Institute under the data transfer agreement. Among the 755 cancer cell lines, 620 (from 279 females and 341 males) with available gender information were used for genotype information analysis in the present study. The details of these cell lines are shown in *Supplementary file 4* . The processed data are in PICNIC output file format, which includes information on genotype, loss of heterozygosity and absolute allelic copy number segmentation (*Greenman et al., 2010*). Greenman et. al. developed the algorithm, PICNIC (Predicting Integral Copy Number In Cancer), to predict absolute allelic copy number variation in cancer (*Greenman et al., 2010*). This algorithm improved the normalization of the data and the determination of the underlying copy number of each segment. It has been used for Affymetrix genome-wide SNP6.0 data from 755 cancer cell lines, which were derived from 32 tissues. The Affymetrix Genome-Wide SNP Array 6.0 has 1.8 million genetic markers, including more than 900,000 single nucleotide polymorphism probes (SNP probes) and more than 900,000 probes for the detection of copy number variation (CN probes).

The genome-wide gene expression data for 947 human cancer cell lines from 36 tumor types were generated by Barretina et al (*Barretina et al., 2012*), as part of the cancer cell line Encyclopedia (CCLE) project using Affymetrix U133 plus 2.0 arrays and are available from the CCLE project website (CCLE_Expression_Entrez_2012-09-29.gct, http://www.broadinstitute.org/ccle/home). The expression profiles of 768 cell lines with gender information, representing 337 females and 431 males, were used in this study. These cell lines were partially overlapped with the lines used in Greenman et. al. Additionally, RNA-seq data from 41 lymphoblastoid cell lines from 17 females and 24 males were downloaded from GEO database (GSE16921) (*Cheung et al., 2010*). The details of these cell lines are shown in *Supplementary file 5*.

## LOH detection and copy number estimation

Human genomes harbor single nucleoid polymorphisms (SNPs) at a density of about 0.5–1 SNP per kb. When a large segment of chromosome is lost in somatic cells, the corresponding region would be devoid of SNPs, referred to as loss of heterozygosity (LOH). LOH regions may regain the copy number but the lost heterozygosity cannot be regained.

We used the genotype information and the allelic copy number estimation generated from PICNIC to infer LOH as well as copy number of a specific chromosome. As for a chromosome, if $\geq$ 95% of SNP sites were homologous we considered that there was a LOH (loss of heterogeneity) event for this chromosome. Similarly, if $\geq$ 95% of detected alleles on the chromosome had a constant copy number of 0, 1, 2, 3 or 4, the copy number would be considered as the copy number of the chromosome. The copy number of the Y chromosome was estimated separately. In females, although all sites on Y chromosome should have yielded 0 copies, only ~ 60% of sites detected by the Y chromosome probes showed a copy number of 0. This result indicated that several X homologous regions on the Y were covered by ~ 30% of Y probes. Therefore, Y chromosome loss was defined as when more than 60% of SNP probes from the Y chromosome showed a copy number of 0.

## Sex chromosome genotype inference

The expression level of XIST can be used as a proxy to distinguish the active X chromosome from the silent one as this gene was expressed on the inactive X chromosome and functioned in cis (*Richardson et al., 2006*). According to Greenman's and Barretina's studies, 496 cancer cell lines have both copy number and expression data. As expected, XIST was silenced in male cell lines, as well as in females with whole X chromosome LOH (*Figure 1C*). Based on X chromosome LOH and copy number information, we identified five genotypes, including XaO (female lines with one X = 20 lines), XaXa (female lines with isodisomy of X = 17 lines), XaXb (female lines with heterozygous for the X = 28 lines), Xa[Y] (male lines with one X = 53 lines) and XaXa[Y] (male lines with two X's = 69 lines).

## C(Xa:A)(ratio of active X's to autosomes) calculation

All male (341 lines) and female cell lines with whole X chromosome LOH (103 lines) were employed for C(Xa:A) calculation. C(Xa:A) was defined as the ratio of absolute X copy number to that of all autosomes.

## $E_{X/A}$ (ratio of X to autosomal expression) calculation

$E_{X/A}$ was defined as the ratio of the expression of X-linked genes to that of autosomal ones. The median values of expressed X-linked and autosomal genes were used to calculate $E_{X/A}$ in both cancerous and normal cell lines. For the datasets from the Affymetrix U133 + 2.0 array, genes with signal intensities $\geq$ 32 ($\log_2 \geq$ 5) were considered to be expressed. While as for RNA-seq data, genes with RPKM values $\geq$ 1 were considered to be expressed.

Previous studies have shown that $E_{X/A}$ value may be affected by gene set used (*Deng et al., 2011*). In addition, several silent genes in normal tissues have been shown to be expressed in tumor tissues (*Hofmann et al., 2008*). Those genes were dominant on X chromosome, which could result in an increase of $E_{X/A}$. To exclude the possibility that $E_{X/A}$ ratios may be biased in cancerous cell lines, gene sets for $E_{X/A}$ calculation were first selected in normal cell lines by three criteria, with the same sets then selected in cancerous cell lines. The three filtering criteria for gene set selection were RPKM > 0, 1, and 5 in normal cell lines (*Figure 2C*).

## Differences in X-linked gene expression between Xa[Y] and XaXa[Y] lines

To explore the impact of extra X chromosome on gene expression levels of X-linked genes, 53 cell lines with Xa[Y] and expression data, 69 cell lines with XaXa[Y] and expression data were used. T-test with Benjamini and Hochberg adjusting method was employed to determine genes, the expression of which are significantly changed due to an extra X copy. 648 detected X-linked genes are plotted in *Figure 2A*. The free statistical programming language R was used for the statistical analysis (version 3.0.1).

## X chromosome Fluorescence in situ hybridization

HeLa cells (from the Culture Collection of the Chinese Academy of Sciences, Shanghai, China) were cultured in DMEM (Life Technologies ,CA, United State) supplemented with 10% fetal bovine serum (FBS), 100 U/ml of penicillin, and 100 μg/ml of streptomycin. A549 cells (from ATCC) were cultured in RPMI-1640 (Life Technologies) with 10% fetal bovine serum (FBS), 100 U/ml of penicillin, and 100 μg/ml of streptomycin at 37°C with 5% CO2. Approximately $2 \times 10^6$ cells were seeded and cultured in 10 cm dishes with 10 ml growth medium as described above. To synchronize the cells, 200 μl of thymidine (100 mM) was added to the cells. After incubating for 14 hr, the cells were washed twice with 10 ml PBS and then supplemented with 10 ml growth medium containing deoxycytidine (24 μM). After incubating for 2 hr, 10 μl nocodazole (100 μg/ml) was added to the cells. The cells were incubated for an additional 10 hr.

After synchronization, cells were harvested and treated with 4 ml hypotonic solution (75 mM, KCl) pre-warmed to 37°C for 30 min. The cells were then fixed via three immersions in fresh fixative solution (3:1 methanol:acetic acid) (15 min each time). The fixed cell suspension was spotted onto a clean microscope slide and allowed to air dry. We used the ''XCyting Chromosome Paints'' and 'Xcyting Centromere Enumeration Probe' (MetaSystems, Germany) for whole X chromosomes and centromere of X chromosome fluorescence in situ hybridization (FISH) analysis, respectively. Following the manufacturer's instructions, 10 μl of probe mixture was added to the prepared slide. The slide was then covered with $22 \times 22$ mm$^2$ cover slip and sealed with rubber cement. Next, the slide was heated at 75°C for 2 min on a hotplate to denature the sample and probes simultaneously, followed by incubation in a humidified chamber at 37°C overnight for hybridization. After hybridization, the slide was washed in 0.4 x SSC (pH 7.0) at 72°C for 2 min, then in 2 x SSC and 0.05% Tween-20 (pH 7.0) at room temperature for 30 s, before being rinsed briefly in distilled water to avoid crystal formation. The slide was drained and allowed to air dry. Finally, 5 μl DAPI (MetaSystems) was applied to the hybridization region and covered with a coverslip. The slide was processed and captured using fluorescence microscopy as recommended (Olympus FV1000, 100X objective). The number of Xs were counted for each individual cell. A total of 343 HeLa cells and 170 A549 cells were screened.

The identification of HeLa cells was confirmed by genome sequence method and the identification of A549 cells was confirmed by karyotype profile. The mycoplasma contamination status was tested by DNA staining for both HeLa and A549.

## The assumption of the model

In the model, autosomes are treated as an integrated set, labeled 'A' and counted as a set. There may be two reasons to do so. One is mechanistic if the entire haploid set of chromosomes increases a unit. While this may happen in organismal evolution, we consider the mechanism dubious for cell lines. In the absence of meiosis, whole-sale changes should involve the entire diploid set (diploids, tetraploids and octoploids, as in human hepatocytes).

We therefore suggest that chromosomes are gained and lost individually. They evolve more or less as a cohesive set in the long run thanks to natural selection that imposes a cost on uneven sets. For autosomes, the dynamics is portrayed in *Figure 4B–C*. When autosomes are gained, say from $n_A = 2$ to $n_A = 3$ ($n_A$ being the number of autosomal haploid set), the imbalance within the autosomal set appears to be tolerated only to a point. Let $\Delta_{ij}$ (i, j = 1, 2,. . 13 for largest 13 autosomes) designate the difference in the number between autosome i and autosome j. In a balanced set, all $\Delta_{ij} = 0$ which represents a fitness peak when all autosomes have the same number. During evolution, $\Delta_{ij} = 1$, having a reduced fitness, can be tolerated but not $\Delta_{ij} >= 2$. (The constraint appears to be loosened for the smaller autosomes.) Thus, the autosomal set evolves between integers of $n_A = 1$–3, with the occasional nA = 4 (see *Figure 4D*). Obviously, moving nA from one whole number to the next is a slow process. In *Equation (1)* of the main text, u represents the change in the number of X and v represents the change in $n_A$. In testing the model, we let u = 10 v but the results are not sensitive to the ratio.

Assuming all genotypes have whole numbers of X and A, we assign a fitness to all genotypes of *Figure 4A*. Under neutrality, these genotypes have the same fitness. The main goal of the modeling work (see *Figure 5B and C*) is to test the fitness neutrality of these 'whole number' genotypes.

## Environmental factors

When we attribute the observed genetic changes associated with cells' unicellular existence, we do include all environmental factors that make the unicellular existence possible. Without these factors, cell lines cannot live. An analogy is the study of the evolution of social structure, which is also conditional on many environmental factors (e.g., food supply) but one often uses 'social structure' as an all-encompassing term. Since the unicellular existence requires a number of environmental factors (which the cell culture community has been keen to identify), it is not possible to separate 'unicellularty' and the environments needed to sustain the unicellular existence. It is also important to point out that these environmental factors are often antagonistic to the multicellular living.

## Strength of selection

In this study, we use the model to compare the observations with the neutral expectation. Although we could reject the neutral model and conclude the direction of selection, we refrain from estimating the strength of selection for two reasons – both biological and technical.

First, the most important demonstration is that the wildtype C(Xa:A)=1:2 is not the fittest genotype. We believe this conclusion in itself is very novel because all evolutionary theories posit the wildtype to be at a local fitness optimum. We could conclude that the wild type C(Xa:A)=1:2 is less fit than C(Xa:A)=2:3 based on *Figure 5B*.

Second, while we could conclude C(Xa:A)=1:2 to be less fit than C(Xa:A)=2:3, estimating the strength of selection is an entirely different proposition. In this case, the main unknown is vT in *Figure 5B and C*. In other words, we do not know how close each cell line is to the equilibrium. Given the various histories of these cell lines, we suspect that the value may range between 0.1 and 10. The differences are qualitatively consistent but the actual values will require knowing the precise culture history of each cell line.

# Acknowledgements

We thank Jian Lu and Xionglei He for comments on an earlier version of the manuscript. Additionally, we thank Yang Shen, Yong E Zhang, Rui Zhang and Wenfeng Qian for numerous constructive discussions.

# Additional information

## Funding

| Funder | Grant reference number | Author |
|---|---|---|
| National Key Basic Research Program of China | 973 Project, 2014CB542006 | Chung-I Wu |
| Strategic Priority Research Program of the Chinese Academy of Sciences | XDB13040300 | Xuemei Lu Chung-I Wu |
| National Science Foundation of China | 31730046 | Chung-I Wu |
| National Science Foundation of China | 91531305 | Xuemei Lu |
| National Key Basic Research Program of China | 985 Project, 33000-18821105 | Chung-I Wu |

The funders had no role in study design, data collection and interpretation, or the decision to submit the work for publication.

## Author contributions

Jin Xu, Conceptualization, Data curation, Supervision, Funding acquisition, Investigation, Writing—original draft, Project administration, Writing—review and editing; Xinxin Peng, Yuxin Chen, Resources, Data curation, Formal analysis, Investigation, Methodology, Writing—original draft, Writing—review and editing; Yuezheng Zhang, Data curation, Formal analysis, Investigation, Methodology,

Writing—original draft, Writing—review and editing; Qin Ma, Resources, Data curation, Writing—review and editing, Acquisition of data; Liang Liang, Ava C Carter, Resources, Data curation, Formal analysis, Methodology, Writing—review and editing; Xuemei Lu, Conceptualization, Formal analysis, Writing—review and editing; Chung-I Wu, Conceptualization, Supervision, Funding acquisition, Investigation, Project administration, Writing—review and editing

### Author ORCIDs

Jin Xu  http://orcid.org/0000-0003-0944-9835
Chung-I Wu  http://orcid.org/0000-0001-7263-4238

### Decision letter and Author response

Decision letter https://doi.org/10.7554/eLife.28070.026
Author response https://doi.org/10.7554/eLife.28070.027

## Additional files

### Supplementary files

• Supplementary file 1. XIST expression in human B-cell, HeLa and A549. The expression of XIST was detected by Real-Time PCR (see note). The $\triangle\triangle Ct$ values were 12 and 11 higher in HeLa and A549 than B-cell, which illustrates nearly $2^{10}$ lower expression of XIST gene in HeLa and A549. The primers of XIST and control gene, used for Real-Time PCR are listed.
DOI: https://doi.org/10.7554/eLife.28070.013

• Supplementary file 2. Copy number of 13 largest autosomes in 6 HeLa sublines measured by low coverage whole genome sequencing. Observation and Poission expectation of G3 is shown in *Figure 4B*, and E7 is shown in *Figure 4C*. Copy number of other small autosomes is also shown.
DOI: https://doi.org/10.7554/eLife.28070.014

• Supplementary file 3. The information of cancer cell lines.
DOI: https://doi.org/10.7554/eLife.28070.015

• Supplementary file 4. The information of non-cancer cell lines.
DOI: https://doi.org/10.7554/eLife.28070.016

• Supplementary file 5. 4-state model that incorporates selection as presented in *Figure 5A*.
DOI: https://doi.org/10.7554/eLife.28070.017

• Transparent reporting form
DOI: https://doi.org/10.7554/eLife.28070.018

### Major datasets

The following previously published datasets were used:

| Author(s) | Year | Dataset title | Dataset URL | Database, license, and accessibility information |
|---|---|---|---|---|
| Barretina J, Caponigro G, Stransky N, Venkatesan K, et al | 2012 | SNP and Expression data from the Cancer Cell Line Encyclopedia (CCLE) Organism Homo sapiens | https://www.ncbi.nlm.nih.gov/geo/query/acc.cgi?acc=GSE36139 | Publicly available at the NCBI Gene Expression Omnibus (accession no: GSE36139) |
| Greenman, et.al | 2010 | COSMIC Cell Lines project | http://grch37-cancer.sanger.ac.uk/cell_lines/download | Available with registration/login at the Catalogue Of Somatic Mutations In Cancer (http://cancer.sanger.ac.uk/cell_lines) |
| Cheung VG, Nayak RR, Wang IX, Elwyn S, et al | 2010 | Polymorphic cis- and trans-regulation of human gene expression | https://www.ncbi.nlm.nih.gov/geo/query/acc.cgi?acc=GSE16921 | Publicly available at the NCBI Gene Expression Omnibus (accession no: GSE16921) |

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
