## [Decision Letter]

Thank you for submitting your article "Free-living human cells reconfigure their chromosomes in the evolution back to uni-cellularity" for consideration by *eLife*. Your article has been reviewed by two peer reviewers, and the evaluation has been overseen by Diethard Tautz as the Reviewing and Senior Editor. The following individual involved in review of your submission has agreed to reveal his identity: Harmit S Malik.

The reviewers have discussed the reviews with one another and the Reviewing Editor has drafted this decision to help you prepare a revised submission.

Summary:

This is a highly provocative manuscript that argues that the chromosomal competition of cells in multicellular organisms is selectively constrained (X:A = 1:2) from achieving what would be a fitness optimum were these cells allowed to be selected purely for their unicellular survival (X:A = 2:3). This convergence is highly unexpected given the divergent sources of the various cell lines analyzed and given that they each carry different driver mutations. Although only a few instances are characterized at the karyotypic level by FISH or chromosomal paints, the SNP array based conclusions are highly significant and unexpected.

However, in spite of the potentially very interesting finding, the reviewers felt that a much clearer presentation is needed in some places. In fact, because of the partial lack of detail in the present manuscript, it is not yet clear whether it can be accepted after these details and clarifications are made. Hence the revised version will need re-review.

Most of the concerns stem from the very cursory style of writing of the manuscript. Please note that the imposed word limit is not necessarily strict, but only a recommendation. Alternatively, some more extended descriptions of details could be shifted to supplementary material. Generally it is necessary to do a better explanation of the methods and the rationale, and most importantly, the caveats.

Essential Revisions:

1) What you refer to as non-cancer cell lines have in fact been immortalized either by telomerase or EBV; they are thus poor surrogates for 'normal cells' but closer to normal than cancer cells that have already undergone karyotypic changes. A table listing all the cell lines is required, including a discussion of the associated caveats.

2) What is the rationale for the expectation that X chromosomes should only be lost at 15%- if this is based on size, this can be better explained using Figure 1.

3) The models that invoke or do not invoke selection can be made more useful where there are means to derive selective coefficients. It appears this is what you headed towards with the transition model, but falls short and instead just concludes that selection must play a role to arrive at a X:A= 2:3 ratio. (see also point 10 below).

4) How have all the cells been cultured? If under serum free or serum rich conditions, then does this factor explain some of the divergence (spread) in the X:A ratios observed. Table 1 that lists the cell lines could make this clear, and this will also allow falsification of the hypothesis that in fact the cells are convergently evolving towards serum-rich media conditions rather than to 'unicellularity' (generically speaking). Sparse description of media and growth conditions is one of the glaring omissions in the present manuscript.

5) It is not acceptable that a critical piece of the data from the HeLa cells is from reference to another unpublished manuscript. At the very least this manuscript should be provided to the reviewers, or the relevant data included in this manuscript (see also point 8 below).

6) The assumption that autosomes are gained individually seems unjustified (even if it’s only for the sake of argument) – more likely failure to undergo proper chromosome segregation led to ploidy changes following individual autosomes might have been lost. While the rationale to only focus on large autosomes seems reasonable, data on the smaller autosomes and their variance should be made available to the reader even if only in figure supplements.

7) The descriptions of the female cell lines are a bit confusing. The general statement or conclusion is that for the female lines with additional X chromosomes, the active (Xa) chromosome is gained. This is important because gains of the inactive X chromosome (silenced) should not confer much selection. However, in Figure 1, XIST expression (associated with X-chromosome inactivation) is observed in most (N=103) cell lines. Also, in Figure 1, only 20% of the female cell lines are illustrated – what is the status of the other 80% of the cell lines? More clarity is needed for evaluation.

8) The HeLa cell line studies are a bit confusing. Figure 2 is used to state that "within-cell heterogeneity does not seem to undermine our conclusions". Then 6 single HeLa cells were isolated and grown to a million cells (subsection “The concerted evolution of autosomes as a set”, second paragraph). WGS was used to evaluate ploidy, with the conclusion that larger (chr 1-13) evolve "cohesively" as a set (see the aforementioned subsection, last paragraph). This is a very interesting observation and conclusion, but the numbers (N=6) are small. However, if true, it should be possible to karyotype single cells to see if the expected 1:2, 2:3 X:A ratios are observed, as predicted. This data is already available (Figure 2) and could just be organized and presented. This data might be already presented in Figure 4—figure supplement 1 and Figure 4—figure supplement 2, but there are male and female peaks in it Figure 4—figure supplement 2, but Figure 4—figure supplement 1 is about the HeLa (female) cell line, so the sources of the male peaks are uncertain. Clarification is needed.

9) The reasoning for Figure 4 should also account for a potential bias in the sense that for the X;A ratios, typically X = 1 or 2 chromosomes and A = 22 or more chromosomes. That X can change when a single chromosome changes ploidy, but A requires 22 chromosomes to change uniformly, a less likely scenario which should be discussed or clarified.

10) Along these lines, the modeling in Figure 5 illustrates "A" as a single entity with integer moves. Given the large number of autosomal chromosomes, the feasibility of this model should be further clarified.

Overall, the X:A ratios are complicated because although one can calculate an average "A" value, the complex rearranged nature of many cancer cell chromosomes (where chromosomes may be hybrids of many different autosomes) can render the meaning of "A" a bit uncertain. It would be useful to present exactly how many cell lines are actually X:A1:2 versus 2:3 versus neither.

---

## [Author Response]

Essential Revisions:1) What you refer to as non-cancer cell lines have in fact been immortalized either by telomerase or EBV; they are thus poor surrogates for 'normal cells' but closer to normal than cancer cells that have already undergone karyotypic changes. A table listing all the cell lines is required, including a discussion of the associated caveats.

We address the issue further in the beginning of Discussion. Indeed, “normal cell lines” means “cell lines derived from non-cancerous cells”. While they are not truly normal (cell lines, by definition, are not normal), they may serve as a benchmark against which the evolution in cancer cell lines can be measured.

We added two supplementary tables, which include the information requested. Supplementary file 4 lists the information of cancer cell lines (subsection “Data collection”, second paragraph). Supplementary file 5 lists the information of non-cancer cell lines (see the aforementioned subsection, last paragraph).

2) What is the rationale for the expectation that X chromosomes should only be lost at 15%- if this is based on size, this can be better explained using Figure 1.

Yes, the expectation of Y/X chromosome loss is calculated from the regression of LOH percentage on chromosome size.

Revision: “Given its rank as the 7th largest chromosome, X is not expected to be lost in more than 15% of cell lines based on the regression analysis of Figure 1”

3) The models that invoke or do not invoke selection can be made more useful where there are means to derive selective coefficients. It appears this is what you headed towards with the transition model, but falls short and instead just concludes that selection must play a role to arrive at a X:A= 2:3 ratio. (see also point 10 below).

The point deserves a bit more elaboration. We added a new section in Materials and methods.

Revision: The following description is reproduced from the new Materials and methods.

“In this study, we use the model to compare the observations with the neutral expectation. Although we could reject the neutral model and conclude the direction of selection, we refrain from estimating the strength of selection for two reasons – both biological and technical. […] Given the various histories of these cell lines, we suspect that the value may range between 0.1 and 10. The differences are qualitatively consistent but the actual values will require knowing the precise culture history of each cell line.”

4) How have all the cells been cultured? If under serum free or serum rich conditions, then does this factor explain some of the divergence (spread) in the X:A ratios observed. Table 1 that lists the cell lines could make this clear, and this will also allow falsification of the hypothesis that in fact the cells are convergently evolving towards serum-rich media conditions rather than to 'unicellularity' (generically speaking). Sparse description of media and growth conditions is one of the glaring omissions in the present manuscript.

The question is very reasonable. Since it is generally true that a limited number of recipes are used to culture cells, how did we conclude that it is “unicellularity”, rather than “culture condition”, that drives the cell line evolution. This question deserves to be answered fully (see below).

Revision: In Materials and methods, we provide a fuller answer, which is reproduced below.

“When we attribute the observed genetic changes associated with cells’ unicellular existence, we do include all environmental factors that make the unicellular existence possible. […] It is also important to point out that these environmental factors are often antagonistic to the multicellular living.”

5) It is not acceptable that a critical piece of the data from the HeLa cells is from reference to another unpublished manuscript. At the very least this manuscript should be provided to the reviewers, or the relevant data included in this manuscript (see also point 8 below).

The study is currently under review but has been posted on BioRxiv, so it is accessible to the general readership (https://www.biorxiv.org/content/early/2017/10/05/193482).

6) The assumption that autosomes are gained individually seems unjustified (even if it’s only for the sake of argument) – more likely failure to undergo proper chromosome segregation led to ploidy changes following individual autosomes might have been lost.

Comment 6 is later expanded and contained in comment 9 and comment 10. We will therefore answer the 3 questions together when answering comments 9-10.

While the rationale to only focus on large autosomes seems reasonable, data on the smaller autosomes and their variance should be made available to the reader even if only in figure supplements.

Copy number of chromosomes smaller than chr13 has been added into Supplementary file 2.

7) The descriptions of the female cell lines are a bit confusing. The general statement or conclusion is that for the female lines with additional X chromosomes, the active (Xa) chromosome is gained. This is important because gains of the inactive X chromosome (silenced) should not confer much selection. However, in Figure 1, XIST expression (associated with X-chromosome inactivation) is observed in most (N=103) cell lines. Also, in Figure 1, only 20% of the female cell lines are illustrated – what is the status of the other 80% of the cell lines? More clarity is needed for evaluation.

We agree that the descriptions and Figure 1 are confusing. The new description, reproduced below, should clarify the issue.

Revision: Figure 1 have been revised as follows:

“We use only female lines that show LOH of the whole X chromosome (~37% of female lines) in counting Xa’s for the following reason. […] Much like male lines of Figure 1, Figure 1 also shows roughly half of female lines to have gained an extra Xa.”

And the figure legends have also been revised.

8) The HeLa cell line studies are a bit confusing. Figure 2 is used to state that "within-cell heterogeneity does not seem to undermine our conclusions". Then 6 single HeLa cells were isolated and grown to a million cells (subsection “The concerted evolution of autosomes as a set”, second paragraph). WGS was used to evaluate ploidy, with the conclusion that larger (chr 1-13) evolve "cohesively" as a set (see the aforementioned subsection, last paragraph). This is a very interesting observation and conclusion, but the numbers (N=6) are small.

Figure 2 and Figure 4 in fact present two related but somewhat different aspects of the evolution. Both suggest the within-line variation shows substantial cohesiveness that is more so than expected. The difference is that Figure 2 examines the variation within lines (by means of FISH focusing on X) and Figure 4 shows the average changes in evolution (by means of DNA sequencing, mainly on the autosomes). Figure 2 is like the study of human-to-human variation and Figure 4 looks into the average changes between humans and chimpanzees.

We agree that N=6 is a bit small but it has turned out to be adequate. Because the underlying Poisson distribution is broad, each line in fact provides more than modest statistical support. In aggregate, there is sufficient power to reject the null model.

However, if true, it should be possible to karyotype single cells to see if the expected 1:2, 2:3 X:A ratios are observed, as predicted. This data is already available (Figure 2) and could just be organized and presented.

We use FISH, coupled with X-specific staining, to count X’s of each individual cell. Counting autosomes of a single cell, on the other hand, is tricky because autosomes are not individually labeled. Some autosomes may be outside of the field or may come from the neighboring cells. In the analysis, we still have to average over cells of the same cell line. In that case, DNA sequencing or old-fashioned microarrays are more accurate in providing the average X:A ratio within a line. (We also know that the variation within a line is modest.)

This data might be already presented in Figure 4—figure supplement 1 and Figure 4—figure supplement 2, but there are male and female peaks in it Figure 4—figure supplement 2, but Figure 4—figure supplement 1 is about the HeLa (female) cell line, so the sources of the male peaks are uncertain. Clarification is needed.

Figure 4—figure supplement 2 presents the counts from multiple cell lines (not HeLa). That is why there are male and female lines.

The next three questions form a set:

[(6) is repeated here: The assumption that autosomes are gained individually seems unjustified (even if it’s only for the sake of argument)- more likely failure to undergo proper chromosome segregation led to ploidy changes following individual autosomes might have been lost.]

9) The reasoning for Figure 4 should also account for a potential bias in the sense that for the X;A ratios, typically X = 1 or 2 chromosomes and A = 22 or more chromosomes. That X can change when a single chromosome changes ploidy, but A requires 22 chromosomes to change uniformly, a less likely scenario which should be discussed or clarified.10) Along these lines, the modeling in Figure 5 illustrates "A" as a single entity with integer moves. Given the large number of autosomal chromosomes, the feasibility of this model should be further clarified.

Following the reviewers’ suggestions, we clarify the description in the expanded Results subsection “A model for the evolution of free-living cells”.

Revision:

As shown in Materials and methods.

“The assumption of the model

In the model, autosomes are treated as an integrated set, labeled “A” and counted as a set. There may be two reasons to do so. […] The main goal of the modeling work (see Figure 5) is to test the fitness neutrality of these “whole number” genotypes.”

Overall, the X:A ratios are complicated because although one can calculate an average "A" value, the complex rearranged nature of many cancer cell chromosomes (where chromosomes may be hybrids of many different autosomes) can render the meaning of "A" a bit uncertain. It would be useful to present exactly how many cell lines are actually X:A1:2 versus 2:3 versus neither.

In the answer to Q7, we lay out the procedure. That is why fewer than half of the lines could be used as we had to discard those lines with partial aneuploids (polyploidy for parts of the chromosomes). We should note that translocations among autosomes with low aneuploidy are all right for the analysis. On the other hand, X: autosome translocations (which are not common) cannot be used.